# Manzano: A Simple and Scalable Unified Multimodal Model with a Hybrid Vision Tokenizer

Yanghao Li[∘†] Rui Qian[∘] Bowen Pan[∘] Haotian Zhang[∘‡] Haoshuo Huang[∘] Bowen Zhang[∘‡]
Jialing Tong[⋆] Haoxuan You[⋆‡] Xianzhi Du[⋆] Zhe Gan[⋆] Hyunjik Kim[⋆‡] Chao Jia[⋆‡] Zhenbang
Wang[⋆] Yinfei Yang[⋆‡] Mingfei Gao Zi-Yi Dou[‡] Wenze Hu Chang Gao Dongxu Li Philipp
Dufter Zirui Wang[‡] Guoli Yin Zhengdong Zhang Chen Chen Yang Zhao Ruoming Pang[‡]
Zhifeng Chen

Apple

[∘] First authors    [⋆] Core authors    [†] Project lead    [‡] Work done at Apple

## Abstract

Unified multimodal Large Language Models (LLMs) that can both understand and
generate visual content hold immense potential. However, existing open-source
models often suffer from a performance trade-off between these capabilities. We
present Manzano, a simple and scalable unified framework that substantially re-
duces this tension by coupling a hybrid image tokenizer with a well-curated training
recipe. A single shared vision encoder feeds two lightweight adapters that produce
continuous embeddings for image-to-text understanding and discrete tokens for
text-to-image generation within a common semantic space. A unified autoregres-
sive LLM predicts high-level semantics in the form of text and image tokens, with
an auxiliary diffusion decoder subsequently translating the image tokens into pixels.
The architecture, together with a unified training recipe over understanding and
generation data, enables scalable joint learning of both capabilities. Manzano
achieves state-of-the-art results among unified models, and is competitive with
specialist models, particularly on text-rich evaluation. Our studies show minimal
task conflicts and consistent gains from scaling model size, validating our design
choice of a hybrid tokenizer.

## 1 Introduction

Unified multimodal models (OpenAI, 2025; Deng et al., 2025; Chen et al., 2025d; Wu et al., 2025a;
Zhou et al., 2024), which integrate both understanding and generation capabilities, have become
increasingly prominent within the research community. The appeal of this paradigm stems from the
discovery that integrating these domains unlocks emergent capabilities (OpenAI, 2025; Deng et al.,
2025) in generation, such as complex world reasoning, multimodal instruction following, and iterative
visual editing. Yet, in practice, adding generation often degrades understanding. Existing unified
models (Deng et al., 2025; Fan et al., 2025; Liang et al., 2024; Chen et al., 2025d) consistently lag
far behind their understanding-only counterparts (Seed, 2025; Bai et al., 2025; Zhang et al., 2024a),
especially on text-rich benchmarks (Mathew et al., 2021; Masry et al., 2022).

A key reason for this gap is the conflicting nature of visual tokenization. Auto-regressive generation
usually prefers discrete image tokens (Chameleon, 2024; Wu et al., 2024; Ma et al., 2025) while
understanding typically benefits from continuous embeddings. Many models adopt a dual-tokenizer
strategy (Wu et al., 2025b; Chen et al., 2025d; Fan et al., 2025; Tong et al., 2024b), using a semantic
encoder for rich, continuous features while a separate quantized tokenizer like VQ-VAE (Van
Den Oord et al., 2017) handles generation. However, this forces the language model to process
two different image token types, one from high-level semantic space versus one from low-level
spatial space, creating a significant task conflict. While some solutions like Mixture-of-Transformers
(MoT) (Liang et al., 2024; Deng et al., 2025) can mitigate this by dedicating separate pathways for
each task, they are parameter-inefficient and are often incompatible with modern Mixture-of-Experts
(MoE) (Fedus et al., 2022; Lepikhin et al., 2021) architectures. An alternative line of work bypasses
this conflict by freezing a pre-trained multimodal LLM and connecting it to a diffusion decoder (Pan
et al., 2025; Wu et al., 2025a;c). While this preserves the understanding capability, it decouples

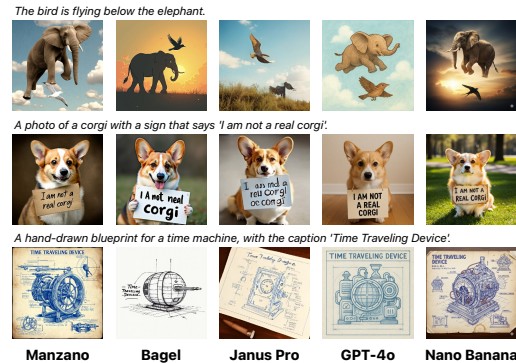
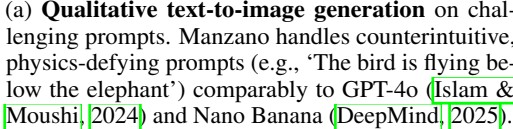
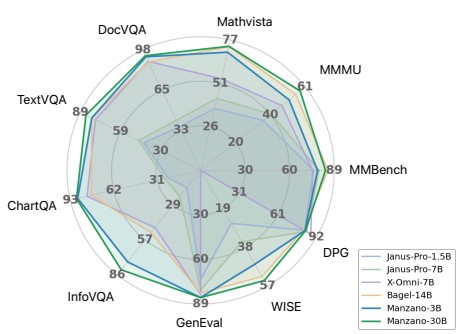

(a) **Qualitative text-to-image generation** on challenging prompts. Manzano handles counterintuitive, physics-defying prompts (e.g., 'The bird is flying below the elephant') comparably to GPT-4o (Islam & Moushi, 2024) and Nano Banana (DeepMind, 2025).

(b) **Quantitative comparisons** on popular understanding and generation benchmarks. Manzano 3B and 30B models achieve superior or competitive performance compared to other SOTA unified multimodal LLMs.

Figure 1: Comparison of qualitative and quantitative results for Manzano model.

generation, losing potential mutual benefits and limiting potential gains for generation from scaling the multimodal LLM.

To overcome the above challenges, we propose **Manzano**, a simple unified model that harmonizes the representations for understanding and generation. Manzano employs a unified shared visual encoder with two lightweight and specialized adapters: a *continuous* adapter for understanding tasks and a *discrete* adapter for generation. Because two adaptors originate from the same encoder, it yields hybrid representations from a homogeneous source, significantly mitigating task conflict in the LLM. We first pre-train the hybrid tokenizer with a small LLM decoder to pre-align the image features with the LLM feature space. Then the autoregressive multimodal LLM is jointly trained on a mixture of pure text, image understanding, and image generation data. Finally, we leverage a diffusion image decoder (Peebles & Xie, 2023; Chen et al., 2025a) to render pixels by taking the generated image tokens as conditioning. By delegating pixel-level synthesis to the diffusion decoder, the LLM can focus more on modeling semantics through an unified autoregressive objective.

We train the unified multimodal LLM with a joint recipe to learn image understanding and generation simultaneously. This training consists of three stages: a pre-training stage on a large-scale corpus of text-only, interleaved image-text, image-to-text (IT), and text-to-image (TI) data; a continued pre-training stage on higher-quality IT and TI data; and a supervised fine-tuning (SFT) stage on curated text, IT, and TI instruction data to enhance instruction following capability and improve both understanding and generation tasks. Each stage serves a distinct role: pre-training on web-scale data builds broad cross-modal alignment, continued pre-training on curated data strengthens domain-specific capabilities such as document and chart understanding, and SFT teaches the model to follow diverse user instructions for both understanding and generation.

We demonstrate that Manzano achieves state-of-the-art performance on both understanding and generation tasks. As shown in Fig. 1a and 1b, our 3B model, despite its smaller LLM size, achieves competitive generation performance compared to other unified multimodal LLMs. Simultaneously, it delivers significantly better understanding performance, especially on text-rich benchmarks that demand precise perceptual capabilities. Our ablations on the training recipes also indicate minimal cross-task conflict under joint training (Fig. 3). These findings suggest that the architecture and the training recipe effectively mitigate the conflict between understanding and generation, even in a compact model.

Facilitated by the simplicity of the architecture and the joint training recipe, we further investigate the *scaling* behavior of Manzano. Our scaling studies in Sec. 4.3 show substantial improvements across both understanding and generation benchmarks when scaling the LLM decoder (from 300M to 30B). In addition, enlarging the diffusion decoder also leads to significant gains in image structural integrity, as validated by large-scale human evaluations.

## 2 RELATED WORK

### 2.1 MLLMs FOR IMAGE UNDERSTANDING

Recent advances in Multimodal Large Language Models (MLLMs) have led to a widely adopted architectural pattern that links a vision encoder with a language model through a trainable interface. Typical vision encoders include CLIP (Radford et al., 2021a), SigLIP (Zhai et al., 2023), and the recent InternViT (Chen et al., 2024c). Early works experimented with elaborate connector designs—for example, Flamingo (Alayrac et al., 2022) incorporates gated cross-attention layers to inject image features into the LLM, while BLIP-2 (Li et al., 2023b) introduces the Q-Former to better align visual and textual representations. A notable departure from these complex strategies is LLaVA (Liu et al., 2023; 2024a), which demonstrates that a lightweight Multi-Layer Perceptron (MLP) projection can effectively serve as the connector. This simplicity has since become the blueprint for many follow-up systems, such as the MM1 series (McKinzie et al., 2024; Zhang et al., 2024a), the InternVL family (Chen et al., 2024c; Zhu et al., 2025; Wang et al., 2025a), and the Qwen-VL models (team, 2024; Bai et al., 2025), which further improve performance by scaling up both data and backbone models. However, these MLLMs are primarily designed for understanding tasks and lack the capability to generate high-quality images, which limits their applicability in tasks that require bidirectional visual–text reasoning and creation. Despite being limited to understanding tasks, MLLMs still provide valuable strengths—their training recipes and scaling strategies are much more mature than those of current unified multimodal models, which we discuss next.

### 2.2 UNIFIED MULTIMODAL MODELS

The integration of image understanding and generation within a single, unified multimodal LLM is becoming prominent. GPT-4o (OpenAI, 2025) demonstrates embedding image generation capabilities directly into an autoregressive LLM, which unlocks emergent abilities, such as stronger instruction following, improved text rendering, multi-turn visual editing, and sophisticated world knowledge reasoning. Existing unified models can be broadly categorized into three architectural paradigms. First, the unified autoregressive (AR) approach (OpenAI, 2025; Chen et al., 2025d;c;b; Han et al., 2025b; Tian et al., 2024; Chameleon, 2024; Tong et al., 2024b; Fan et al., 2025; Ma et al., 2025; Han et al., 2025a; Geng et al., 2025; Wu et al., 2025d; 2024; Wang et al., 2025c) converts images into sequences of discrete or continuous tokens, enabling LLM to jointly model both image and text sequences in an autoregressive manner. Second, the decoupled LLM-diffusion approach (Pan et al., 2025; Wu et al., 2025a;c) employs a largely frozen LLM for semantic understanding and contextual reasoning, while delegating image synthesis to a separate diffusion decoder. In this design, the LLM itself does not possess native image generation capability. Third, the hybrid AR-diffusion approach (Deng et al., 2025; Zhou et al., 2024; Liang et al., 2024) integrates both paradigms within a single transformer, using autoregressive decoding for text and an embedded diffusion process for images. Our model is most closely aligned with the first, autoregressive paradigm. However, instead of employing separate tokenizers (Deng et al., 2025; Chen et al., 2025d; Fan et al., 2025; Wu et al., 2025c) for understanding and generation, we introduce a unified semantic tokenizer to produce both continuous features for understanding tasks and quantized features for generation tasks. This hybrid tokenizer strategy substantially mitigates the task conflict that commonly arises. Moreover, while our LLM backbone follows the autoregressive design, we augment it with a diffusion decoder for image synthesis, enabling high-fidelity generation guided by the semantic representations.

### 2.3 DIFFUSION MODELS FOR IMAGE GENERATION

Diffusion-based generative models (Song et al., 2020; Ho et al., 2020; Dhariwal & Nichol, 2021) have become one of the most prominent approaches for high-fidelity image synthesis. These models gradually refine Gaussian noise into realistic images through a learned denoising process. Latent diffusion methods (Rombach et al., 2022; Podell et al., 2023) enhance computational efficiency by conducting generation in the latent space of a pre-trained variational autoencoder (VAE) (Kingma & Welling, 2022), reducing memory and compute requirements while preserving visual quality. More recently, flow matching approaches (Liu et al., 2022; Ma et al., 2024; Tong et al., 2023) have been introduced to connect source and target distributions via simplified continuous trajectories, leading to further gains in synthesis performance (Esser et al., 2024). In parallel, architectural advances such as Diffusion Transformers (DiTs) (Peebles & Xie, 2023; Chen et al., 2024a; Esser et al., 2024; Labs,

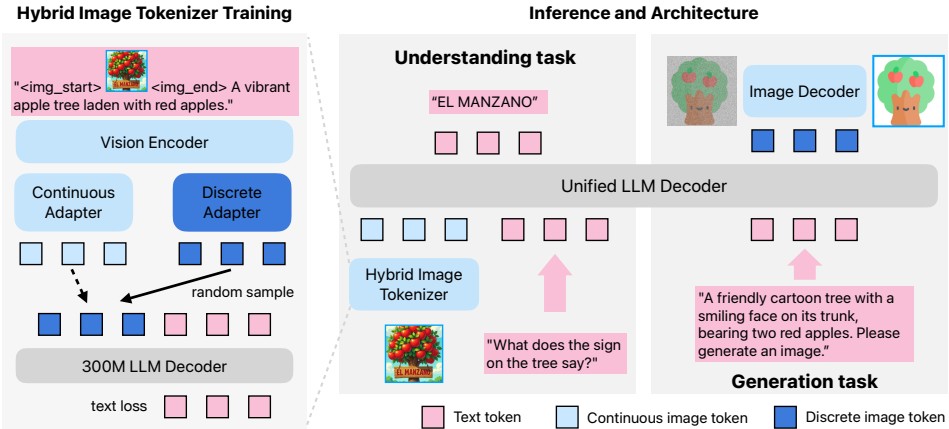

Figure 2: **Our hybrid tokenizer workflow**. **(Left)**: The tokenizer produces two distinct but homogeneous feature streams through separate adapters. During training, one adapter output is randomly sampled and passed to a small LLM decoder for alignment. **(Right)**: Once the tokenizer is trained, the right panel illustrates how these two feature types are applied to understanding and generation tasks.

2024; Chen et al., 2025a) have demonstrated strong scalability and quality improvements, echoing the success of transformer-based designs in natural language processing.

Building on these developments, our diffusion decoder integrates the strengths of the field by employing a DiT in the latent domain with a conditional flow matching objective. Unlike conventional text-to-image diffusion models (Ramesh et al., 2022; Saharia et al., 2022) conditioned on semantic embeddings from pre-trained text encoders such as CLIP (Radford et al., 2021a), our method leverages visual token embeddings generated by the LLM as conditioning signals.

## 3 MODEL

Manzano is a multimodal large language model (MLLM) that unifies understanding and generation tasks using the auto-regressive (AR) approach. The architecture comprises three components: ($i$) a *hybrid vision tokenizer* that produces both continuous and discrete visual representations; ($ii$) an *LLM decoder* that accepts text tokens and/or continuous image embeddings and auto-regressively predicts the next discrete image or text tokens from a joint vocabulary; and ($iii$) an *image decoder* that renders image pixels from predicted image tokens (see Figure 2 for the framework).

### 3.1 DESIGN CHOICES

**Unified hybrid representation.** The hybrid image tokenizer encodes images into continuous tokens for understanding (I2T), and discrete tokens for generation (T2I), while sharing the same visual encoder: (i) *Continuous for I2T*: Manzano utilizes continuous embeddings for I2T tasks, a strategy widely adopted in popular visual understanding models (team, 2024; Seed, 2025), which has shown superior performance, especially on text-rich tasks that require more visual details (e.g., DocVQA, ChartQA, and InfoVQA). Our ablation (Table 1) also shows discrete tokens underperform on understanding tasks, which reflects the weak understanding results reported for some pure-discrete unified models (Chameleon, 2024; Wang et al., 2024). (ii) *Discrete for T2I*: Representing images as discrete code indices lets the LLM use the same AR next-token learning strategy as text, simplifying the generation pipeline and scaling behavior. (iii) *Shared unified semantic space*: Both branches originate from the same encoder backbone; thus, continuous and discrete tokens inhabit a common semantic space, which reduces potential task conflict. The LLM decoder focuses on regressing high-level semantics (text and image tokens), while the diffusion decoder is responsible for rendering high-fidelity details in pixel space. Many existing unified models rely on separate tokenizers for understanding and generation (Chen et al., 2025d; Deng et al., 2025) — for instance, using a CLIP tokenizer for understanding tasks and a VAE tokenizer for generation. Although this strategy preserves more image spatial details, it exacerbates the task conflict within the subsequent LLM.

Some studies (Chen et al., 2025b;c) find that a dedicated generation tokenizer is not as compatible with LLM as the semantic tokenizer. Thus, our hybrid unified image tokenizer employs a single image encoder for both understanding and generation tasks.

**Simplicity and scalability.** Our design keeps the training losses standard and components cleanly decoupled, which simplifies unification and scaling for the unified MLLM in these aspects: (i) *Unified AR objective*: Our unified LLM decoder uses a single AR objective for text-only, I2T, and T2I tasks without additional auxiliary losses or per-task heads. (ii) *Decoupled components*: The clear split between semantic prediction (LLM decoder) and detail generation (image decoder) supports independent scaling of the base LLM and the image decoder. (iii) *Practical scaling*: Our approach readily leverages mature, scalable training pipelines from LLM/MLLM and diffusion decoders. By contrast, prior works (e.g., Transfusion (Zhou et al., 2024) and Bagel (Deng et al., 2025)) explored incorporating auto-regressive text prediction and a diffusion image generation process for image generation in one LLM, but left large-scale scaling under-explored. Our decoupled design facilitates scaling the LLM to 30B and diffusion decoder to 3B, yielding promising scaling behavior (Sec. 4.3).

## 3.2 ARCHITECTURE

**Hybrid Image Tokenizer.** Our tokenizer comprises three components: ($i$) a standard vision transformer (ViT) (Dosovitskiy et al., 2020) as the vision backbone; ($ii$) a continuous adapter, which first applies a $3 \times 3$ Spatial-to-Channel (STC) layer to reduce the number of spatial tokens by a factor of 9 (e.g., from $42 \times 42 \times 1024$ to $14 \times 14 \times 9216$) and then uses an MLP to project each feature into the LLM feature dimension (e.g., 2048); and ($iii$) a discrete adapter, which also starts with the STC compression step but further quantizes the features using finite scalar quantization (FSQ) (Mentzer et al., 2023) — chosen for its simplicity and scalability to large codebooks (64K in our experiments) — before applying an MLP projection into the LLM feature dimension.

**Unified LLM.** We connect our hybrid image tokenizer to a standard text LLM decoder for unified training on a mixture of datasets containing text, understanding, and generation data. For the language backbone, we leverage internal pre-trained LLMs.

**Image Decoder.** We train an image decoder on top of a pre-trained hybrid image tokenizer to reconstruct images in pixel space from discrete image tokens. Given an input image, the hybrid tokenizer first encodes it into a latent representation, which serves as the conditioning input for a flow-matching pipeline (Lipman et al., 2022) that transports Gaussian noise into realistic images. For the decoder backbone, we adopt the DiT-Air architecture (Chen et al., 2025a), which employs a layer-wise parameter-sharing strategy that reduces the size of the standard MMDiT model (Esser et al., 2024) by approximately 66% while maintaining comparable performance. We provide three decoder configurations with the parameter sizes of 0.9B, 1.75B, and 3.52B, supporting a range of output canvas resolutions from 256 to 2048 pixels.

**Inference Pipeline.** The inference pipeline for both understanding and generation tasks is shown in Fig. 2 (right). For understanding tasks, Manzano uses the hybrid image tokenizer to extract continuous features. These features, along with text features, are then fed into the unified LLM decoder to predict the final answer. For generation tasks, Manzano takes a text input and predicts a sequence of image tokens. The image decoder then renders these tokens into image pixels.

## 4 EXPERIMENTS

In this section, we first introduce the evaluation setup for understanding and generation capabilities (Sec. 4.1). We then study the cross-token interplay in our model (Sec. 4.2). After that, we analyze the scaling behavior by varying model sizes (Sec. 4.3). Finally, we compare our model against state-of-the-art models, including both specialist and unified models. The training details, including the data mixture for understanding and generation, can be found in Appendix A.

## 4.1 EVALUATION

We evaluate our models on image understanding and generation capabilities on popular benchmarks.

| Tokenizer Paradigm | Understanding Tasks | | | Generation Tasks | | |
|---|---|---|---|---|---|---|
| | General | Knowledge | Text-Rich | GenEval | DPG | WISE |
| Pure-Discrete | 63.3 | 62.2 | 62.3 | 77 | 80.9 | 35 |
| Dual-Encoder | 63.8 | 63.6 | 72.0 | 65 | 66.3 | 17 |
| Hybrid Tokenizer | 64.9 | 66.5 | 73.3 | 77 | 79.9 | 35 |

Table 1: **Tokenizer strategy ablation.** Tokenizers are evaluated with a 1B unified LLM model. The full list of evaluation tasks for understanding can be referred to Sec. 4.4.1. The hybrid tokenizer outperforms the other two tokenizer paradigms.

**Understanding.** We adopt the three categories of benchmarks for multimodal understanding: (i) *General VQA*: SeedBench (Li et al., 2023a), RealWorldQA (Zhang et al., 2024b), and MMBench (Liu et al., 2024b). (ii) *Knowledge & Reasoning*: AI2D (Kembhavi et al., 2016), ScienceQA (Lu et al., 2022), MMMU (Yue et al., 2023), and MathVista (Lu et al., 2023). (iii) *Text-rich Document & Chart Understanding*: ChartQA (Masry et al., 2022), TextVQA (Singh et al., 2019), DocVQA (Mathew et al., 2021), InfoVQA (Mathew et al., 2022), and OCRBench (Liu et al., 2024c).

**Generation.** We use both automated and human evaluations: (i) *Automated Evaluation*: The automated benchmarks include GenEval (Ghosh et al., 2023) and DPGBench (Hu et al., 2024) for prompt following generation, and WISE (Niu et al., 2025) for World Knowledge-Informed generation. (ii) *Human Evaluation*: We curate a comprehensive evaluation set comprising 800 challenging prompts, subsampled from established academic benchmarks (Wiles et al., 2025; Yu et al., 2022) and from widely used community evaluation platforms. The generated outputs are assessed by in-house human raters on three dimensions: structural integrity, instruction following, and aesthetic quality. For each dimension, raters assign one of three grades: major issues, minor issues, or no issues, and are quantized to scores afterwards. To mitigate bias, entity information is masked, and the sample order is randomized. Each sample is independently rated by three raters, and the final scores are obtained by averaging across raters to reduce variability.

## 4.2 UNDERSTANDING-GENERATION INTERPLAY

In this section, we study the task conflict along two axes: (i) *tokenizer strategy* (pure-discrete vs. dual-encoder vs. our hybrid); (ii) *task mixing* (unified vs. single-task). For simplicity, we skip the continued pre-training stage in the unified LLM training for these ablations.

**Tokenizer Strategy.** We construct two baselines to compare our unified hybrid tokenizer strategy: (i) *Pure-discrete.* Prior works (Chameleon, 2024; Wang et al., 2024; Wu et al., 2024) train a quantized semantic vision tokenizer using various quantization techniques (Mentzer et al., 2023; Van Den Oord et al., 2017) and then use an LLM to predict the next text and image tokens. To mimic these methods in our setting, we replace the understanding inputs for LLM with discrete features from our hybrid tokenizer, so the LLM uses the same discrete tokens for both understanding and generation. To isolate the effect of quantization on understanding, we use the same weights for the vision encoder and the discrete adapter from our hybrid tokenizer. (ii) *Dual-encoder.* Other popular models (Chen et al., 2025d; Deng et al., 2025) use a dual-encoder strategy to preserve detailed features by a semantic encoder for understanding and a VAE-style encoder for generation, effectively mitigating the degradation of understanding. We reproduce this baseline by replacing the discrete tokens from our hybrid tokenizer with those generated by an internal reproduction of MagViT-2 (Yu et al., 2023), an autoencoder-style tokenizer. This MagViT-2 tokenizer uses FSQ (Mentzer et al., 2023) with a 64K codebook and a spatial compression ratio of 8. For generation tasks, we resize images to 128x128 pixels instead of the original 256x256. This reduced the number of tokens per image to 256, which we found improved the model's instruction-following capabilities on benchmarks.

Table 1 shows the results on both image understanding and generation tasks. Our hybrid tokenizer paradigm shows the least task conflict and outperforms both pure-discrete and dual-encoder baselines on all tasks. Specifically, the pure-discrete baseline leads to a significant drop in understanding performance–especially on text-rich benchmarks, due to information loss from quantization. While the dual-encoder baseline mitigates some of this degradation, it still consistently underperforms our hybrid tokenizer on all understanding tasks–especially on knowledge benchmarks, which rely heavily

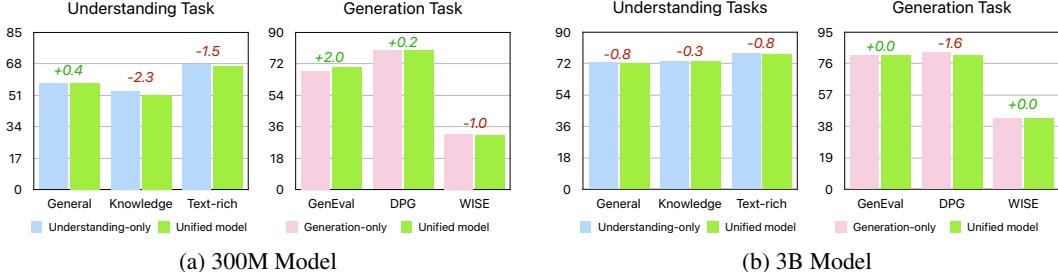

(a) 300M Model                                           (b) 3B Model

Figure 3: **Unified *vs*. Single-task study.** Our unified model exhibits a slight regression compared with the understanding-only model on understanding tasks; however, this effect becomes negligible at the 3B scale, where the gap is less than 1.0. For generation, the unified model shows a decline on only one benchmark compared with the generation-only model.

on the LLM's reasoning abilities. This suggests that the conflict between heterogeneous visual tokens resides within the LLM.

**Unified vs. Single-task.** To quantify the task conflict in our hybrid tokenizer paradigm, we compare our unified model with baselines trained exclusively for understanding or generation. For the understanding-only baseline, we remove all text-to-image data from both the pre-training and SFT stages. We reduce the training steps to ensure it is exposed to the same number of text and image understanding tokens as our unified model. Similarly, for the generation-only baseline, we remove the understanding data and keep only the text-only and text-to-image data, while also reducing the training steps. We conduct this ablation study with a 300M and a 3B LLM decoder. The results, plotted in Fig. 3a and 3b, show that the unified LLM trained with our hybrid tokenizer performs on par with the dedicated, single-task models on nearly all tasks, even at a compact size like 300M. This demonstrates that our unified hybrid tokenizer paradigm successfully unifies visual perception and generation without a performance trade-off.

## 4.3    MODEL SCALING BEHAVIOR

Facilitated by the decoupled design of LLM Decoder and Image Decoder, we explore the model scaling behavior along two dimensions: *LLM Decoder* and *Image Decoder*. Similar to Sec. 4.2, we skip the continued pre-train stage in the unified LLM training for the scaling experiments.

**Scaling LLM Decoder.**    We vary only the LLM Decoder size (300M, 1B, 3B, and 30B) while keeping the image decoder (0.9B), data mixtures, and training hyperparameters fixed[1]. Fig. 4a shows monotonic gains across all understanding (General / Knowledge / Text-Rich) and generation (GenEval / DPG / WISE) metrics as the LLM decoder scales. Compared to 300M, our 3B Manzano model improves significantly by +14.2 (General), +18.8 (Knowledge), +10.9 (Text-rich), +11.0 (GenEval), +1.48 (DPG), +12.0 (WISE). Further scaling to 30B yields smaller but consistent gains over 3B. Fig. 7 shows the qualitative examples for image generation. We can see that the generation capabilities, including instruction-following, text-rendering, and overall image quality, are improved consistently across different LLM scales. The results support the simple yet effective design for Manzano: LLM decoder captures high-level semantics, and scaling it benefits both understanding and generation.

**Scaling Image Decoder.**    We evaluate the performance of image decoders of varying sizes built on top of a 3B LLM decoder. Figure 4b shows that, in human evaluations, structural integrity improves substantially (+9.9), while instruction following performance remains unchanged. A slight decrease is observed in aesthetic quality. For automatic generation benchmarks, performance on GenEval and DPGEval remains nearly identical, whereas WISE exhibits a modest improvement (+2.0).

**Takeaways.**    Scaling the unified LLM backbone consistently improves both understanding and generation, with substantial gains on text-rich understanding tasks and on WISE for generation. Scaling the image decoder also enhances image quality, without negatively affecting understanding.

---

[1]We pre-train 30B LLM Decoder on roughly half the tokens compared to other model sizes.

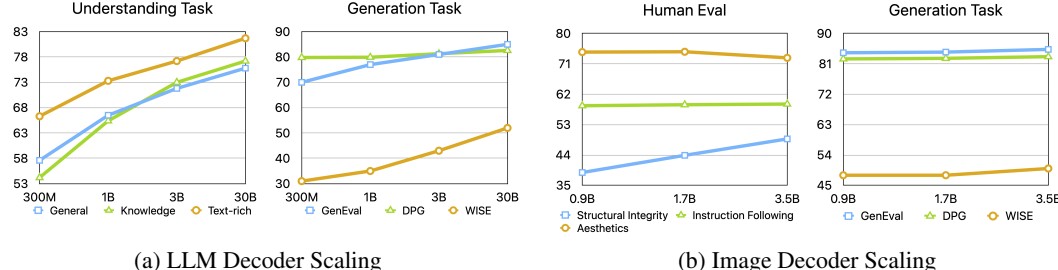

(a) LLM Decoder Scaling                    (b) Image Decoder Scaling

Figure 4: **Model scaling behavior of Manzano**. (a) Scaling the LLM decoder yields monotonic improvements across both understanding and generation benchmarks. (b) Scaling the image decoder enhances structural integrity while maintaining stable quantitative benchmarks. A drop in aesthetic quality is observed, which we leave for more in-depth study in future work.

| Model | General Benchmarks | | | Knowledge Benchmarks | | | | Text-rich Benchmarks | | | | |
| | SEED[I] | RealWorldQA | MMBench (dev-en) | AI2D (test) | SQA (test) | MMMU (val) | MathV (testmini) | ChartQA (test) | TextVQA (val) | DocVQA (test) | InfoVQA (test) | OCRBench (test) |
|---|---|---|---|---|---|---|---|---|---|---|---|---|
| *3B Specialist Model* | | | | | | | | | | | | |
| BLIP-3-4B (Xue et al., 2024) | 72.2 | 60.5 | – | – | 88.3 | 41.1 | 39.6 | – | 71.0 | – | – | – |
| Phi-3-Vision-4B (Abdin et al., 2024) | 71.8 | 59.4 | 78.6 | 76.7 | 90.8 | 40.4 | 44.5 | 81.4 | 70.1 | 83.3 | 49.0 | 63.7 |
| MM1.5-3B (Zhang et al., 2024a) | 72.4 | 56.9 | 72.4 | 65.7 | 85.8 | 37.1 | 44.4 | 74.2 | 76.5 | 87.7 | 58.5 | 65.7 |
| InternVL2.5-2B (Chen et al., 2024b) | – | 60.1 | 77.2 | 74.9 | – | 43.6 | 51.3 | 79.2 | 74.3 | 88.7 | 60.9 | 80.4 |
| InternVL2.5-4B (Chen et al., 2024b) | – | 64.3 | 78.7 | 81.4 | – | 52.3 | 60.5 | 84.0 | 76.8 | 91.6 | 72.1 | 82.8 |
| InternVL3.5-4B (Wang et al., 2025b) | – | 66.3 | – | 82.6 | – | 66.6 | 77.1 | 86.0 | 77.9 | 92.4 | 78.0 | 82.2 |
| Qwen2.5VL-3B (Bai et al., 2025) | – | 65.4 | 76.4 | 81.6 | – | 53.1 | 62.3 | 84.0 | 79.3 | 93.9 | 77.1 | 79.7 |
| *30B Specialist Model* | | | | | | | | | | | | |
| LLaVA-NeXT-34B (Liu et al., 2024a) | 75.9 | – | – | 81.8 | 51.1 | 46.5 | – | 69.5 | – | – | – | |
| Cambrian-34B (Tong et al., 2024a) | 75.3 | 67.8 | – | 79.7 | 85.6 | 49.7 | 53.2 | 75.6 | 76.7 | 75.5 | – | 60.0 |
| MM1.5-30B (Zhang et al., 2024a) | 75.0 | 69.0 | – | 77.2 | 91.9 | 47.4 | 55.6 | 83.6 | 79.2 | 91.4 | 67.3 | 65.8 |
| InternVL2.5-26B (Chen et al., 2024b) | – | 74.5 | – | 86.4 | – | 51.8 | 67.7 | 87.2 | 82.4 | 94.0 | 79.8 | 85.2 |
| *Unified Model* | | | | | | | | | | | | |
| Blip-3o-4B (Chen et al., 2025b) | 73.8 | 60.4 | 78.6 | – | 46.6 | – | – | 78.0 | – | – | – | |
| Emu3-8B (Wang et al., 2024) | 68.2 | 57.4 | 58.5 | 70.0 | – | 31.6 | 47.6 | – | 64.7 | 76.3 | – | 68.7 |
| Janus-Pro-7B (Chen et al., 2025d) | 72.1 | – | 79.2 | 68.1[†] | – | 41.0 | 42.5 | 25.8[†] | 45.6[†] | 40.8[†] | 21.3[†] | 59.0[†] |
| X-Omni-7B (Geng et al., 2025) | 74.1 | 62.6[†] | 74.8 | 76.8[†] | – | 47.2[†] | 54.1[†] | 81.5[†] | 77.4[†] | 88.6 | 46.9[†] | 70.4[†] |
| Bagel-14B (Deng et al., 2025) | 78.5[†] | 72.8[†] | 85.0[†] | 89.2[†] | – | 55.3 | 73.1 | 78.5[†] | 80.0[†] | 88.1[†] | 51.0[†] | 73.3[†] |
| Manzano-3B | 74.3 | 65.1 | 78.1 | 82.2 | 92.9 | 51.4 | 69.8 | 88.2 | 80.1 | 93.5 | 75.0 | 85.7 |
| Manzano-30B | 76.0 | 70.1 | 83.4 | 86.0 | 96.2 | 57.8 | 73.3 | 89.0 | 84.4 | 94.3 | 81.9 | 86.3 |
| GPT-4o (Islam & Moushi, 2024) | 77.1 | 75.4 | – | 84.6 | 90.7 | 69.2 | 61.3 | 85.7 | – | 92.8 | – | 73.6 |
| Gemini-2.5-Pro (Comanici et al., 2025) | – | 78 | 86.3 | 89.5 | 88.4 | 81.7 | 82.7 | 83.3 | 76.8 | 94.0 | 84.3 | 86.2 |

Table 2: **Comparison on general, knowledge, and text-rich benchmarks**. GPT-4o, Gemini-1.5-Pro, and Gemini-2.5-Pro numbers are from OpenVLM Leaderboard. † represents that the results were reproduced independently in our experiments and might differ from those reported in previous studies. Manzano demonstrate competitive understanding capabilities, especially on text-rich benchmarks.

We observed that performance on the GenEval and DPG benchmarks becomes saturated when the model becomes larger. This saturation motivates a re-examination of how emergent capabilities of unified models could be assessed, as existing benchmarks may capture only a limited portion of overall capability and can be boosted through targeted data tuning (Liu et al., 2025). Meanwhile, we observe substantial improvements on world-knowledge generation tasks, and we hope these findings pave the way for new directions in future community research.

## 4.4 COMPARISONS WITH UNIFIED AND SPECIALIST MODELS

To comprehensively assess our Manzano model's capabilities, we compare its performance against SOTA unified and specialist models (i.e., understanding-only and standalone generation models).

### 4.4.1 IMAGE UNDERSTANDING

As mentioned in Sec. 4.1, we evaluate our model's understanding capabilities from three perspectives: Knowledge & Reasoning, General Visual Question Answering, and Text-rich Document & Chart Understanding. The results, shown in Table 2, compare our model against other understanding-only models of a similar size. Despite being a unified model, our model achieves state-of-the-art performance on many understanding benchmarks: (i) *Knowledge & Reasoning*: At the 3B scale, our model outperforms all unified models within the 7B scale and achieves performance on par with or better than the best specialist models at the 3B size. At the 30B scale, our model ranks

| Model | GenEval Benchmark | | | | | | | WISE Benchmark | | | | | | |
|---|---|---|---|---|---|---|---|---|---|---|---|---|---|---|
| | Single | Two | Counting | Colors | Position | Color Attr. | Overall | Cultural | Time | Space | Biology | Physics | Chemistry | Overall |
| *Dedicated T2I Model* | | | | | | | | | | | | | | |
| SDXL-3.5B (Podell et al., 2023) | 0.98 | 0.74 | 0.39 | 0.85 | 0.15 | 0.23 | 0.55 | 0.43 | 0.48 | 0.47 | 0.44 | 0.45 | 0.27 | 0.43 |
| DALL-E 3 (OpenAI, 2024) | 0.96 | 0.87 | 0.47 | 0.83 | 0.43 | 0.45 | 0.67 | – | – | – | – | – | – | – |
| SD3-Medium-2B (Esser et al., 2024) | 0.99 | 0.94 | 0.72 | 0.89 | 0.33 | 0.60 | 0.74 | 0.43 | 0.50 | 0.52 | 0.41 | 0.53 | 0.33 | 0.45 |
| PixArt-Alpha-0.6B (Chen et al., 2023) | – | – | – | – | – | – | – | 0.45 | 0.50 | 0.48 | 0.49 | 0.56 | 0.34 | 0.47 |
| FLUX.1-dev-12B (Labs, 2024) | 0.98 | 0.93 | 0.75 | 0.93 | 0.68 | 0.65 | 0.82 | 0.48 | 0.58 | 0.62 | 0.42 | 0.51 | 0.35 | 0.50 |
| *LLM & Diffusion Conjunction* | | | | | | | | | | | | | | |
| MetaQuery-XL-7B (Pan et al., 2025) | – | – | – | – | – | – | 0.80† | 0.56 | 0.55 | 0.62 | 0.49 | 0.63 | 0.41 | 0.55 |
| OmniGen2-7B (Wu et al., 2025c) | 1.00 | 0.95 | 0.64 | 0.88 | 0.55 | 0.76 | 0.80 | – | – | – | – | – | – | – |
| Qwen-Image-27B (Wu et al., 2025a) | 0.99 | 0.92 | 0.89 | 0.88 | 0.76 | 0.77 | 0.87 | 0.67 | 0.67 | 0.80 | 0.62 | 0.79 | 0.41 | 0.67 |
| *Unified Multimodal LLM* | | | | | | | | | | | | | | |
| Janus-Pro-7B (Chen et al., 2025d) | 0.99 | 0.89 | 0.59 | 0.90 | 0.79 | 0.66 | 0.80† | 0.30 | 0.37 | 0.49 | 0.36 | 0.42 | 0.26 | 0.35 |
| Bagel-14B-A7B (Deng et al., 2025) | 0.99 | 0.94 | 0.81 | 0.88 | 0.64 | 0.63 | 0.82 | 0.44 | 0.55 | 0.68 | 0.44 | 0.60 | 0.39 | 0.52 |
| X-Omni-7B (Geng et al., 2025) | 0.98 | 0.95 | 0.75 | 0.91 | 0.71 | 0.68 | 0.83† | – | – | – | – | – | – | – |
| Manzano-3B | 0.98 | 0.91 | 0.82 | 0.71 | 0.78 | 0.71 | 0.85 | 0.42 | 0.51 | 0.59 | 0.45 | 0.51 | 0.32 | 0.46 |
| Manzano-30B | 1.00 | 0.91 | 0.83 | 0.87 | 0.84 | 0.65 | 0.85 | 0.58 | 0.50 | 0.65 | 0.50 | 0.55 | 0.32 | 0.54 |
| GPT-4o (Islam & Moushi, 2024) | 0.99 | 0.92 | 0.85 | 0.92 | 0.75 | 0.61 | 0.84 | 0.81 | 0.71 | 0.89 | 0.83 | 0.79 | 0.74 | 0.80 |

Table 3: **Comparison on GenEval and WISE benchmarks**. † represents the evaluation that involves LLM rewriting. Manzano achieves competitive performance compared with other unified models.

first on the ScienceQA, MMMU, and MathVista benchmarks and third on the AI2D benchmark, outperforming all other unified and specialist models in these categories. Notably, our model surpasses the proprietary models listed in the final three rows on ScienceQA and is competitive with the current state-of-the-art model on the AI2D benchmark. (ii) *General Visual Question Answering*: For general visual question answering, our model generally outperforms other unified models, despite its smaller size. It also achieves competitive results with state-of-the-art specialist models at both scales. (iii) *Text-rich Document and Chart Understanding*: On text-rich and chart understanding tasks, our model achieves the best performance on four out of five benchmarks (ChartQA, TextVQA, DocVQA, and OCRBench) when compared to all other unified, specialist, and proprietary models. For the InfoVQA task, our model significantly outperforms its unified counterparts and achieves the best results among specialist models.

### 4.4.2 IMAGE GENERATION

We present the quantitative results for our model's image generation capabilities, evaluating them on two benchmarks: GenEval (Ghosh et al., 2023) and WISE (Niu et al., 2025). While both benchmarks assess how well models follow text instructions, WISE additionally evaluates semantic grounding through world-knowledge-informed attributes. As shown in Table 3, our model achieves SOTA results among unified MLLMs on both GenEval and WISE. The 3B model can already perform competitively with or better than much larger unified models, and scaling to 30B further improves generation quality – most notably yielding a large gain on WISE, while maintaining strong GenEval performance. This confirms that our unified architecture and training recipe support strong instruction-following generation. Furthermore, we show the editing capabilities of our model in Sec. 5.

### 4.4.3 COMPARISON WITH UNIFIED MODELS

In addition to specialist models, we also compare against recent unified models such as Janus-Pro (Chen et al., 2025d), X-Omni (Geng et al., 2025), and Bagel (Deng et al., 2025), which aim to handle both understanding and generation within a single framework. Our Manzano model substantially outperforms these unified baselines across almost all understanding benchmarks. At a similar scale, our 3B model exceeds X-Omni and BAGEL on DocVQA, OCRBench, and SEEDBench while maintaining competitive performance on MathVista and ChartQA. Our 30B model further extends this lead, consistently surpassing all existing unified models across knowledge, general VQA, and text-rich domains. This demonstrates that unification does not have to come at the cost of understanding capability. With careful architectural and training design, our model matches or surpasses the best specialist models while providing strong generative capability. We provide more qualitative comparison to the state-of-the-art unified models in Fig. 8.

## 5 CAPABILITY EXTENSION TO IMAGE EDITING

Image editing represents both a crucial application and a natural extension of text-to-image generation. Despite Manzano demonstrating strong multimodal modeling capabilities, particularly on text-rich understanding benchmarks, achieving pixel-level precision in fine-grained image editing remains challenging. Similarly, recent work within the decoupled LLM–diffusion paradigm (Wu et al., 2025c) reports difficulties when relying solely on the LLM for precise editing, since the LLM lacks native mechanisms for direct pixel-level control.

We adopt an approach similar to (Wu et al., 2025c) by providing the reference image simultaneously to both the LLM and the diffusion decoder. In this formulation, the LLM is responsible for diverse instruction following and maintaining semantic coherence, while the diffusion decoder enforces precise pixel-level control. By jointly conditioning on the reference image, Manzano enables accurate semantic instruction following while preserving fine-grained visual consistency. In Fig. 5, Manzano demonstrates versatile editing capabilities, including instruction-guided editing, style transfer, inpainting, outpainting, and depth estimation.

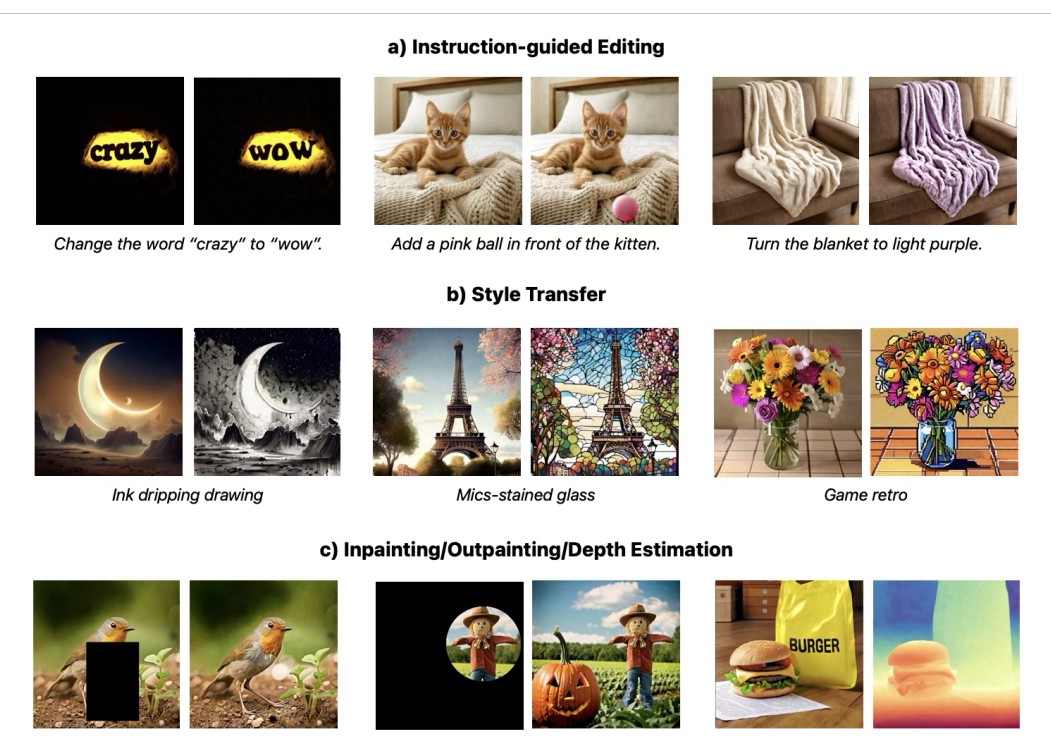

Figure 5: **Editing capabilities of Manzano**. (a) instruction-guided editing, (b) style transfer across diverse visual domains, and (c) extended editing tasks including inpainting, outpainting, and depth-estimation. Manzano achieves pixel-level controls across these five editing tasks.

## 6 CONCLUSION

We introduced Manzano, an MLLM that combines visual understanding and image generation through a hybrid image tokenizer and a unified autoregressive backbone. The LLM predicts high-level semantics in the form of text and image tokens, while a lightweight diffusion-based image decoder renders final pixels from the generated image tokens. Coupled with a streamlined three-stage training recipe, this architecture delivers: ($i$) state-of-the-art on understanding tasks, ($ii$) substantial gains on generation among unified models, and ($iii$) minimal task interference as validated by interplay and scaling ablations. Beyond generation, Manzano naturally supports image editing by conditioning both the LLM and image decoder on a reference image, enabling instruction-following with pixel-level control.

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
