Figure 6: **Training overview**. **(Left)**: Unified LLM training with hybrid tokens, the continuous adapter produces embeddings used for the text loss, while the discrete adapter generates hard tokens serving as targets for the image loss. **(Right)**: With vision encoder and adapters fixed, an image decoder is trained to reconstruct images using a diffusion loss.

## A  TRAINING

### A.1  DATA

Our training data mixture includes text-only, image understanding, and generation data, divided into pre-training, continued pre-training, and supervised fine-tuning (SFT) stages. We leverage high-quality text-only data (Zhou et al., 2025) for both pre-training and SFT to maintain the language modeling capability of Manzano model.

#### A.1.1  PRE-TRAINING & CONTINUED PRE-TRAINING

**Understanding.** We use two types of image understanding data: captioning (paired images and text descriptions), and interleaved image-text data. For captioning, we use a combination of sources with 2.3B image-text pairs, including CC3M (Sharma et al., 2018), CC12M (Changpinyo et al., 2021), COYO (Byeon et al., 2022), VeCap (Lai et al., 2024), and in-house licensed data. This data undergoes a filtering and re-captioning process to ensure high quality. For interleaved data, we use 1.7B documents from (Laurençon et al., 2024) and web-crawled interleaved data, similar to MM1 (McKinzie et al., 2024) and MM1.5 (Zhang et al., 2024a).

In the continued pre-training stage, we further train on 24M high-quality capability-oriented data, including documentation, charts, multilingual OCR, knowledge & reasoning, high-quality synthetic captions data, all with image splitting (Lin et al., 2023; Gao et al., 2024; Zhang et al., 2024a) enabled.

**Generation.** The image generation pre-training data consists of 1B in-house text-to-image pairs. Following (Chen et al., 2025a), we generate synthetic captions using different captioner models. For the continued pre-training stage, we select a high-quality subset of licensed images and re-caption them with a more powerful MLLM, generating descriptions of lengths varying from 20 to 128 tokens.

#### A.1.2  SUPERVISED FINE-TUNING

**Understanding.** Following MM1.5 (Zhang et al., 2024a), our final understanding SFT recipe comprises 75% image–text data and 25% text-only data. The image–text portion is further composed of approximately 30% general knowledge data, 20% document and chart understanding data, and 25% vision chain-of-thought (CoT) and in-house generated reasoning data.

**Generation.** Our text-to-image SFT data includes a curated blend of real and synthetic data. We begin with real-world text-image pairs from the DreamO dataset (Mou et al., 2025). However, we

observed that training solely on this dataset, while sufficient for standard diffusion-based generators, caused our unified auto-regressive model to overfit. To mitigate this, we expand our training data with synthetic examples. First, we incorporated 90K text-image pairs from established datasets, including DALLE3-1M (Egan et al., 2024), BLIP-3o (Chen et al., 2025b), and ShareGPT-4o (Cui et al., 2024). Second, to achieve a larger scale, we generated an additional 4M pairs by feeding prompts from JourneyDB (Sun et al., 2023) into an open-source standalone diffusion model, Flux.1-schnell (Labs, 2024).

## A.2 TRAINING RECIPES

### A.2.1 HYBRID TOKENIZER TRAINING

The hybrid image tokenizer aims to produce two types of tokens: *continuous* for understanding and *discrete* for generation, which are pre-aligned with the multimodal LLM semantic space.

We first pre-train the vision encoder (ViT) using CLIP Radford et al. (2021b). Then we attach a pretrained small LLM decoder (300M) to the shared vision encoder through two parallel continuous and discrete adapters (See Fig. 2-Left). For each training sample, we randomly select one adapter and feed the corresponding embeddings to the LLM decoder, which is trained with next-token prediction. We unfreeze all parameters and train the model on a variety of understanding data domains, including general knowledge, reasoning, and text-rich tasks.

This process enhances the tokenizer's understanding capability, encompassing both high-level semantic understanding and fine-grained spatial details. Meanwhile, the branches are also being aligned to the same space. We follow the pre-training, continued pre-training and SFT stages using the understanding and text-only data described in Sec. A.1.

After training, we discard the small LLM decoder and retain the resulting hybrid image tokenizer, which is then used as a vision input module for the unified LLM and image decoder.

### A.2.2 UNIFIED LLM TRAINING

As shown in Fig. 6-Left, we freeze the parameters of both the vision encoder and the discrete adapter to maintain a fixed vocabulary of image tokens during training. We extend the LLM embedding table with 64K image tokens following the same codebook size of FSQ layer in the tokenizer.

For image understanding, the image tokenizer extracts the continuous features from the input image and feeds them directly into the LLM with standard next-token loss on text targets. For image generation, the tokenizer uses its discrete adapter to convert input images into a sequence of discrete image token IDs that are mapped to image tokens via the extended LLM embedding table. The LLM then computes a cross-entropy loss on these image tokens only. To balance the training of understanding and generation tasks, we set the weight ratio of text loss to image loss at 1:0.5.

We train the unified LLM in three stages. Pre-training and continued pre-training use a 40/40/20 mix of image understanding, image generation and text-only data as described in Sec. A.1.1. We train our model with 1.6T tokens (0.8T tokens for the 30B model) during the pre-training and an additional 83B tokens during the continued pre-training. Similarly, SFT stage uses curated instruction data with a 41/45/14 mix ratio for understanding, generation, and text using datasets in Sec. A.1.2.

### A.2.3 IMAGE DECODER TRAINING

Our image decoder is trained following a progressive resolution growing paradigm (Esser et al., 2024; Chen et al., 2025a). We first pre-train the decoder at a resolution of 256x256 for 400K steps. Subsequently, the model is fine-tuned progressively on higher resolutions of 512, 1024, and 2048, with each stage trained for a shorter schedule of 100K steps. For each stage, only images with short sides larger than the target resolution were used for training.

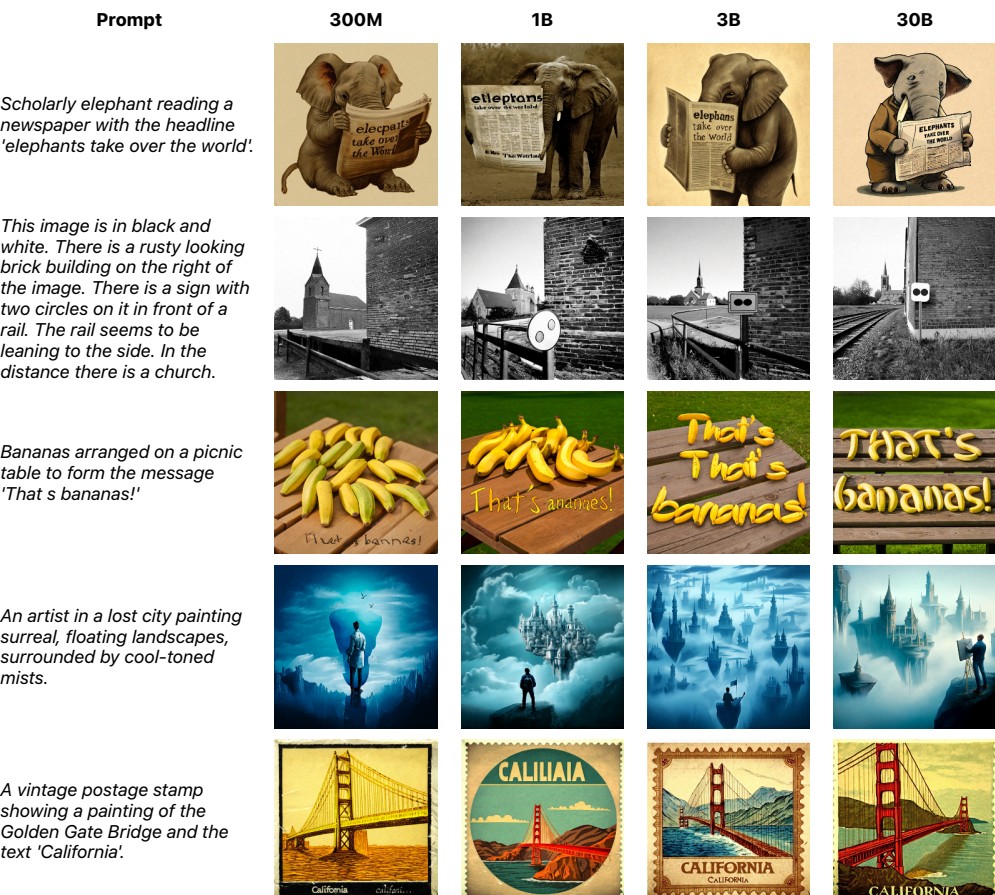

Figure 7: **Qualitative generation results when scaling LLM decoder size**. The generated image quality improves as the LLM decoder size increases. For example, in rows 1, 3, and 5, there is a clear trend toward better text rendering and creativity. In row 2, the scene configuration improves significantly with each increase in the LLM decoder's scale. The 300M model generates an image with only the brick building and the church that are mentioned in the prompt, but as the model grows to 1B and 3B, it begins to include the sign with two circles. Furthermore, the 30B model generates an image that accurately depicts and integrates all the concepts mentioned in the prompt.

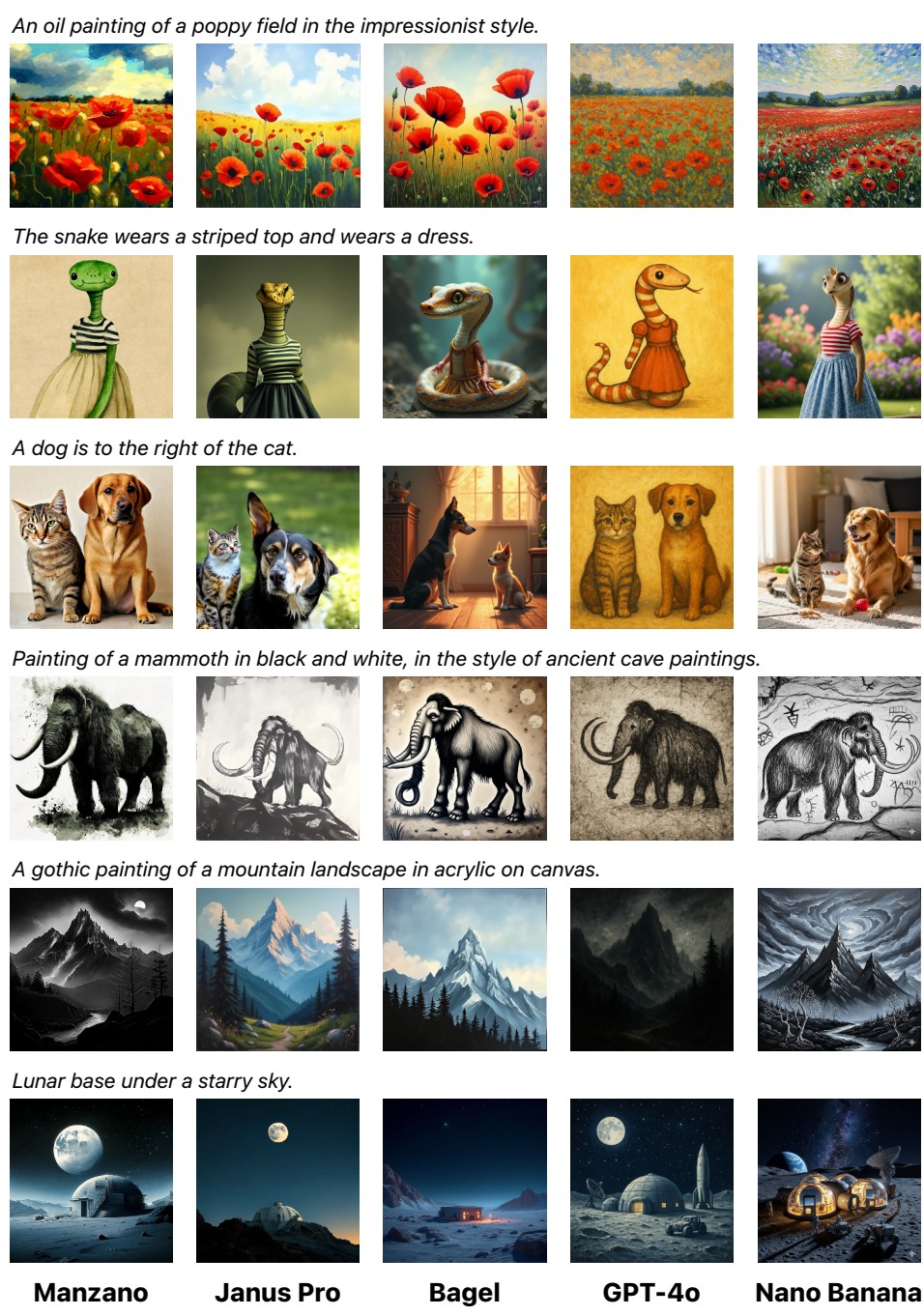

Figure 8: **Qualitative comparison with SOTA unified models**. We compare our Manzano-30B model to the SOTA models through side-by-side comparison. The images generated by our model demonstrate strong capabilities in instruction following, aesthetics, and creativity, often with a photo-realistic quality.