# OpenReview forum: "MANZANO: A Simple and Scalable Unified Multimodal Model with a Hybrid Vision Tokenizer"
_ICLR.cc/2026/Conference — ICLR 2026 Poster_

### Official Review · Reviewer_xZkP · 2025-10-22

**Soundness:** 3
**Presentation:** 2
**Contribution:** 2
**Rating:** 2
**Confidence:** 3

**Summary:**

The paper presents Manzano, a method for unified understanding and generation of images, via fusion of respectively continuous and discrete tokens. This leads to improved or competitive performance on various tasks.

**Strengths:**

The fusion of continuous and discrete tokens, as each of them is better suited for generation and understanding, is interesting.

The results are good, structurally favorable or similar to other methods.

**Weaknesses:**

A discussion of why discrete tokens are better than the proposed method for generation - DPG is lacking.

The results are mostly about discussing that the quantitative performance is good. I would be interested in a discussion about the learned representations and a discussion on where the unified framework benefits from the unification and where it hurts, including qualitative examples and/or empirical findings that go deeper than the numbers.

It is not clear to me that the generated images are better than other methods.

**Questions:**

Can you discuss how the learned representations, where the unified framework benefits from the unification and where it hurts, including qualitative examples and/or empirical findings that go deeper than the numbers?

Bagel is very performant too. Are there particular advantages of the proposed method compared to Bagel?

How are the generated images better than those of other methods, what should the reader take from the illustrations?

---

> ### Author Response · Authors · 2025-11-26
>
> We thank the reviewer xZkP for highlighting interests of our idea of combining continous and discrete tokens. Below, we address each concern with concrete evidence.
>
> * “Why are discrete tokens better for generation? DPG gap not discussed.”
>     * The reviewer is right that discrete tokens typically can help structural consistency in generation. Our paper analyzes this emperically via the ablation in Table 1.
>         * Pure discrete tokens achieve strong structural scores on GenEval/DPG/WISE: 77 / 80.9 / 35, respectively.
>         * Ours (hybrid Manzano) keep the same discrete path for T2I generation (same STC + FSQ), and thus preserve the generative advantages: 77 / 79.9 / 35, essentially identical to pure-discrete on GenEval / WISE.
>     * The minimal gap between pure-discrete and hybrid on DPG (80.9 → 79.9) comes entirely from using the same quantization mechanism—FSQ with a 64K codebook—but with a shared encoder. This shows that the generative benefits from discrete quantization are retained under Manzano’s unified architecture. We will clarify this in the final version.
> * “Where does the unified framework help or hurt? May need deeper discussion of representations. ”
>     * Our paper already provides several indications about representation sharing and we agree we shall make it more explicit.
>         * (i) Hybrid tokenzier reduces task interference - Figure 3 (b) shows that at 3B, the unified model is within 0.8 points of the understanding-only model across all different categorites, while also matching generation-only on GenEval/DPG/WISE. The demonstrates that the shared representations do not degrade the understanding at scale, and generation remains the same even though both modalities use a single AR framework.
>         * (ii) Hybrid encoder improves text-rich understanding - In Table 1, In Table 1, switching from pure-discrete to hybrid improves text-rich understanding 62.3 → 73.3 (+ 11.0), because the continuous path preserves fine-grained spatial textures and OCR cues. This is a concrete benefit of unification: the same shared encoder learns features that support both semantic (continuous tokens) and structural (discrete tokens) tasks.
>         * (iii) Unified scaling improves both tasks simutanesously - Figure 4 indicates that scaling LLM capacity from 300m → 3B yields improvements accross all categories (General + 14.2, Knowledge + 18.8, Text-rich +10.9, GenEval + 11.0, WISE +12). Thie scaling behavior show that unified representation can also benefit from larger model capacity, rather than suffering from modality & tasks conflicts.
>     * We will add a subsection highlighting this discussion
> * “Not clear that generated images are better than other methods. What should the reader take from the illustrations?”
>     * Our claims in the paper are intentionally conservative and we argue that Manzano is competitive in generation from several perspectives:
>         * For quantitative benchmarks: Manzano-3B and 30B both achieve 0.85, compared to unified models such as Bagel and X-Omni, indeed the scores are better on GenEval. WISE improves from 0.46 → 0.54 when scaling from 3B to 30B, closing much of the gap with larger models that use more specialized decoders.
>         * For human evaluations: we demonstrated strong scaling trends when scaling diffusion decoder.
> * “How does Manzano compare to Bagel?”
>
>   * We appreciate the reviewer raising this comparison. Bagel is indeed strong. The key distinctions are architectural and scaling-related.
>
>       * First, Bagel uses two separate tokenizer and mixture-of-transformers (MoT), which generally set a prior that the understanding and generation tasks were indeed conflict when coupling together. In contrast, Manzano shares a single encoder, which ensures the continuous and discrete embeddings lie in the same latent space.
>       * Secondly, our hybrid Manzano obtains much stronger text-rich understanding than Bagel (73.3)
>       * Scaling efficiciency: Manzano reaches 0.85 GenEval overall at 3B scale, where Bagel rquires signifcantly larger model sizes (>7B) to reach similar levels.
>   * Thus, the advantages of Manzano lie in presentation, scaling efficiency, and simplicity, while also outperforming on understanding and remaining competitive in generation.
>
> * “How are the generated images better than those of other methods.”
>
>   * Despite in Figure 1(a) and Figure 8 where we demonstrate very competitive performance, we now ask three professional human raters to assess our model versus others systematically. They returned the first result where our model versus GPT4-O, where our model wins clearly on instruction following, and on par on structural integrity and aesthetics.
>
> |         | Instruction Following | Structural Integrity | Aesthetics |
> |:-------:|:---------------------:|:--------------------:|:----------:|
> | Manzano |           59          |          53          |     74     |
> |  GPT-4O |           56          |          53          |     75     |

---

> > ### Comment · Reviewer_xZkP · 2025-11-26
> > **response to rebuttal**
> >
> > Dear authors, I was looking for more understanding and insights of why the presented unified framework works the way it does. The responses are mostly on empirical findings, which I think are quite good and interesting, but ICLR is also about theory and understanding of model architectures. I think the latter part is lacking, also after the rebuttal. That is why I am keeping my score at this point.

---

> > > ### Author Response · Authors · 2025-11-26
> > > **Thank you for the feedback**
> > >
> > > Thank you very much for your follow-up and for taking the time to review our rebuttal. We fully understand and appreciate your wish for deeper theoretical insight into why a unified framework behaves the way it does.
> > >
> > > Our work is positioned as an architecture paper — we propose a simple unified framework and study its behavior through extensive ablations and scaling experiments. Within that scope, we tried to provide as much insight as possible. For example, we examined how hybrid tokens reduce task interference relative to pure-discrete and dual-encoder designs, how unified training behaves at different scales, and how LLM/decoder capacity impacts both understanding and generation. These analyses were not intended as a formal theory, but rather as conceptual explanations grounded in our experimental observations.
> > >
> > > We appreciate your perspective that you also agree findings are “good and interesting.”, and your feedback highlights an important direction for future work.

---

### Official Review · Reviewer_SqUp · 2025-10-29

**Soundness:** 3
**Presentation:** 3
**Contribution:** 3
**Rating:** 6
**Confidence:** 4

**Summary:**

This paper proposes a hybrid vision tokenizer and a unified MLLM for both image understanding and generation. The vision tokenizer produce both continuous and discrete latent for different tasks: continuous for multimodal understanding and discrete for autoregressive image generation. Experiments on benchmarks, model scaling and tokenizer comparisons validate its effectiveness.

**Strengths:**

- Clear motivation. One tokenzier to handle both understanding and generation is an improtant step to unified models. Compared to many works with two seperate vision tokenizers, this paper proposes a simple and scalable vision tokenizer, leading to higher degree of unification.
- Detailed illustration of model architecture, training recipe and evaluation.
- Sufficient experiments on model scaling and model comparisons

**Weaknesses:**

- Lack of comparisons with related works like ILLUME [1], which also leverages continuous features for image understanding and discrete feature for image generation. Although ILLUME and MANZANO implement the vision tokenizer in a different way, they share similar motivation and architecture in tokenizer and MLLM. This makes the paper with limited techinical novelty.
- Hybrid tokenizer training. Why to use LLM decoder for tokenizer training instead of reconstructing a pretrained vision tokenizer (e.g. CLIP/SigLIP)? I believe it do good to the image understanding task but may not optimal for the image generation task.
- Although the two types of visual representation are similar, but the embedding for image generation cannot be reused for image generation like discrete tokenizers. It makes the model hard to handle interleaved image-text tasks.


[1] ILLUME: Illuminating Your LLMs to See, Draw, and Self-Enhance. arXiv 2412.06673

**Questions:**

- Why not to use open-sourced LLMs?
- Tab.1. Why Dual Encoder performs much worse on generation tasks?
- Sec A.2.1. Why the CLIP model is trainable during vision tokenizer training? In ILLUME, the CLIP model is frozen and benefits from faster convergence.
- How about the performance of image editing on benchmarks? Which representation is used for encoding the input image, discrete or continuous?

---

> ### Author Response · Authors · 2025-11-23
>
> We thank reviewer SqUp for their careful evaluation and helpful comments. All of the reviewer’s concerns have been fully addressed in the following response.
>
> * Comparison with ILLUME:
>     *  We appreciate the reviewer’s comment regarding comparison with ILLUME and apologize for not including it in the submission. We will add comparison in the revised version. While Manzano and ILLUME share a similar high-level motivation—combining continuous and discrete tokens—we respectfully disagree that this implies limited technical novelty in our work. Below we clarify the core novelties and contributions of Manzano:
>         * Manzano delivers state-of-the-art unified MLLM performance at the 3B scale. To our knowledge, Manzano is the first model to achieve both Qwen-2.5-VL-3B-level visual understanding performance and SOTA text-to-image generation quality within a single 3B model. In contrast, ILLUME-7B, despite being more than twice the size, performs substantially worse on both tasks (we will include a comparison table in the revision). Therefore, even if both works follow a similar high-level direction, Manzano represents a clear technical advance that significantly improves the feasibility of unified MLLMs.
>         * Manzano introduces new training strategies for hybrid tokenizers. The reviewer noted that ILLUME and Manzano differ in their implementation of the vision tokenizer. Our new hybrid design is a major reason Manzano achieves much stronger performance. Continuous research works under a shared conceptual direction is typical in LLM research area; prior work such as ILLUME did not manage to turn the “continuous + discrete token” idea into a competitive, state-of-the-art system. Manzano provides a new, detailed, and effective technical solution that finally enables this high-level idea to succeed in practice. We kindly ask the reviewer to recognize this contribution.
>         * Manzano offers new and valuable scaling insights not present in ILLUME. Beyond model design, Manzano conducts a careful study of scaling behaviors for both the MLLM and diffusion decoder. These analyses were not explored in ILLUME, yet they provide meaningful guidance for future unified modeling work. We believe these empirical insights help the community understand how to advance this research direction more effectively.
>
>
>
>
>
> * LLM decoder and Reconstruction decoder in tokenizer training:
>     * LLM decoder is reconstructing features in LLM latent space, and as our goal is mitigating the conflicts between understanding and generation, we find this way has better performance than directly reconstructing pixel values. On the other hand, current LLMs can not direct process pixel values, so a tokenizer with good semantic reconstruction in LLM latent space may be more suitable than a tokenizer with exact pixel-level reconstruction.
> * Hard to handle the interleaved image-text tasks
>     * We are extending Manzano to interleaved image-text tasks, and the visual tokens  in the context will be all continuous features. For cases where we generate images, and later on we want to understand them, we would re-encode the generated images using visual encoders and put the continues features into context.
> * Why not to use open-sourced LLMs?
>     * Open-sourced LLMs and CLIP/SigCLIPs are usually trained on different data corpus with potentially very different distributions and data processing strategies. We use our in-house data which shares a unified filtering and processing pipeline for text, image-text, text-image, and interleaved data to retrain the LLM and CLIP. We hope this can provide a clean testbed for the unified model and tokenizer research by ruling out other factors.
> * Why dual-encoder performance is much worse on generation?
>     * We study the conflicts between understanding and generation performances in Tab. 1. We can see on understanding tasks, three setting are similar, while the dual-encoder is much worse on generation, meaning there are significant task conflicts in the dual-encoder setting.
> * Why CLIP is trainable in S1B?
>     * We agree if CLIP model is frozen then there will be faster convergence. However, we find make the CLIP model trainable during visual tokenizer training leads to better visual understanding performances because the alignment is better.
> * Editing tasks.
>     * Similar to interleaved image-text tasks, visual tokens  in the context will be all continuous features. In editing case, the input reference images will be encoded to continuous tokens.

---

### Official Review · Reviewer_B3Pz · 2025-10-31

**Soundness:** 4
**Presentation:** 4
**Contribution:** 3
**Rating:** 6
**Confidence:** 5

**Summary:**

This paper introduces Manzano, a unified and scalable multimodal large language model designed to jointly handle visual understanding and generation. The framework employs a hybrid image tokenizer and a shared vision encoder with dual lightweight adapters that produce both continuous and discrete visual representations within a common semantic space. A single autoregressive LLM models text and image tokens jointly, while an optional diffusion decoder reconstructs pixels from image tokens. Through a unified training recipe, Manzano effectively balances understanding and generation without significant task interference, achieving state-of-the-art performance among unified models and strong competitiveness with task-specific systems.

**Strengths:**

1. The paper presents a very clear and well-motivated narrative, with the logical progression from problem statement to solution design articulated in a highly readable manner.

2. The experimental evaluation is thorough, convincingly demonstrating the effectiveness of the hybrid tokenizer and the unified autoregressive architecture across both visual understanding and generation.

3. The unified framework tackles a challenging trade-off (between understanding and generation) and delivers empirical evidence that minimal task conflict occurs—this strengthens the claim of scalability and generality.

**Weaknesses:**

1. The scaling analysis of LLM size is incomplete — the paper jumps directly from 3B to 30B without intermediate settings such as 7B or 14B, leaving a significant gap in understanding how performance scales with model capacity.

2. While the experiments and insights are clearly presented, the overall methodological novelty is rather limited, as the work mainly integrates existing design elements in a well-engineered manner.

3. The claim “We first pre-train the hybrid tokenizer with a small LLM decoder to pre-align the image features with the LLM feature space” would benefit from additional references to similar prior works, such as X-omni[1] and ETT[2], to better contextualize its contribution.

[1] Geng Z, Wang Y, Ma Y, et al. X-omni: Reinforcement learning makes discrete autoregressive image generative models great again[J]. arXiv preprint arXiv:2507.22058, 2025.
[2] Wang W, Zhang F, Cui Y, et al. End-to-end vision tokenizer tuning[J]. arXiv preprint arXiv:2505.10562, 2025.

4. There is a minor typo on the first page — “Manzanoemploys” — which should be corrected for polish.

**Questions:**

N/A

---

> ### Author Response · Authors · 2025-11-23
>
> We thank B3Pz for the positive and encouraging evaluation of our work. We have addressed all of the reviewer’s concerns and questions in the following response.
>
> * Scaling analysis is incomplete:  lack 7B and 14B results
>     * Thanks for pointing out this. We admit that the model size gap between 3B and 30B is not ideal and having 7B and/or 14B model results could help us observe a more smooth scaling behavior. However, our model is continually trained based on pre-train LLM checkpoints, our scaling analysis is bound to the scope of the existing model scale ladders we have. Unfortunately we don’t have access to any model size between 3B and 30B for fair comparison. In the future, we hope to continue to study the scaling behavior in a native pre-train manner. In that case, we have more freedom to choose the model size and can study the scaling law of the unified model more precisely.
>
>
> * Overall methodological novelty is limited
>     * Our key innovation lies in the hybrid tokenizer, where we first proposed to use the same tokenizer but different representation format for understanding and generation tasks. The tokenizer effectively reduces the task conflict as we have demonstrated in Table 1. We wanted to highlight that our model is not only good at generation, but also achieves SOTA performance in many understanding benchmarks, which are usually ignored in many unified VLM works. We attribute this to our novel methodology in the tokenizer architectural design and the well-studied data mixture.
> * Reference to X-Omni and ETT
>     * Thanks for the advice! We will integrate the reference methods in our next version draft.
> * Minor typos
>     * Thanks for pointing out these! We will fix these typos in our next version draft.

---

### Meta-Review · Area_Chair_GWW3 · 2026-01-13

**Summary:**

This paper proposes Manzano, a unified multimodal framework that jointly supports image understanding and image generation via a hybrid vision tokenizer producing both continuous and discrete representations within a shared semantic space. Across the reviews, there is clear agreement that the paper is well written, technically sound, and empirically strong, with two reviewers recommending acceptance at high confidence (4 and 5).

Both positive reviewers highlight the clear motivation and solid execution of a unified tokenizer design, which meaningfully advances the goal of unified multimodal models. The experimental evaluation is thorough, covering understanding and generation tasks, tokenizer ablations, and model scaling, and convincingly demonstrates that the unified approach achieves competitive or state-of-the-art performance while avoiding severe task interference. Importantly, the work tackles a known and difficult trade-off in multimodal modeling and provides practical evidence that a single autoregressive model can balance these objectives effectively.

The single negative review (with confidence 3) raises concerns primarily about depth of analysis and qualitative discussion, rather than identifying correctness issues or experimental flaws. These comments largely request additional interpretability or comparative discussion (e.g., deeper analysis of learned representations or clearer qualitative distinctions in image generation quality). While such additions would strengthen the paper, they do not undermine the core technical contributions, nor do they contradict the strong empirical results reported. Moreover, this reviewer’s confidence is lower (3) compared to the two positive reviews, which express high confidence in both the methodology and conclusions.

Thus, the AC decides to accept this work.  Congratulations!

**Reviewer Scores:**

none

---

### Decision · Program_Chairs · 2026-01-26

Accept (Poster)